# Protective Effects of Fermented Glasswort (*Salicornia herbacea* L.) on Aged Gut Induced by D-Galactose in Rats

Doyoung Song [1,†] , Neeracha Sangpreecha [1,†], Saoraya Chanmuang [2], Yang-Kyun Park [1] and Kyung-Sik Ham [1,*]

1   Department of Food Engineering, Solar Salt Research Center, Mokpo National University, Muan 58554, Republic of Korea; doyoung6110@gmail.com (D.S.); jann.n2101@gmail.com (N.S.)
2   Department of Food Science and Technology, Institute of Agriculture and Life Science, Gyeongsang National University, Jinju 52828, Republic of Korea; c.saoraya@gmail.com
*   Correspondence: ksham@mokpo.ac.kr; Tel.: +82-61-450-6275
†   These authors contributed equally to this work.

**Abstract:** Glasswort (*Salicornia herbacea* L.) is a halophyte plant known to contain high contents of minerals and phytochemicals. The purpose of this study was to investigate the effect of fermented glasswort on gut health in D-galactose (D-gal)-induced aging rats. Forty rats were randomly divided into five groups: control (CON), D-gal alone (CD), D-gal plus fructooligosaccharide as a positive control (FOS, 400 mg/kg), D-gal plus low dose fermented glasswort (LGW, 10 mg/kg), and D-gal plus high-dose fermented glasswort (HGW, 20 mg/kg). Each treatment was orally administered to rats of each group for eight weeks. All groups except for the CON group (treated with saline) were intraperitoneally injected with D-gal (150 mg/kg). Our results showed that butyric acid among short-chain fatty acids (SCFAs), goblet cells of colon, and thickness of mucus layer in colon were increased in fermented glasswort groups. In addition, fermented glasswort reduced levels of thiobarbituric acid-reactive substances (TBARS, a related oxidative stress marker) and expression levels of inflammation-related proteins such as IL-1β. These results suggest that fermented glasswort can improve age-related gut health.

**Keywords:** aging; gut health; glasswort; fermentation; goblet cells

## 1. Introduction

Aging refers to a progressive decline in biological structures and functions over time, resulting in increased susceptibility to external stress, disease, and mortality risk. Aging detrimentally affects intestinal function, which is closely associated with a multitude of diseases [1]. The intestine, where approximately 70% of immune cells are gathered, has an important relationship with the immune system. It is constantly damaged by mechanical stress, chemical stress (such as carcinogens from the external environment), and biological stress from harmful bacteria [2]. To maintain intestinal homeostasis, a crucial element is the intestinal barrier upheld by the mucus layer. This layer is composed of mucin produced by goblet cells in the intestinal epithelial layer. It plays vital physiological roles in nutrient absorption, promotion of beneficial gut microbiota growth, inhibition of pathogenic bacteria invasion, immune regulation, and maintenance of intestinal homeostasis [3,4]. Hence, the mucus layer plays a pivotal role in intestinal health. Aging can decrease expression levels of genes associated with intestinal mucus, resulting in decreased thickness of the intestinal mucosa and alterations in gut microbiota compositions [5]. Recent studies have revealed a correlation between aging and reduced diversity within core microbiota groups as well as decreased levels of short-chain fatty acids (SCFAs) [6]. Metabolites such as SCFAs produced by gut microbiota through dietary fiber fermentation are crucial for maintaining gut homeostasis and reducing health risk [7]. In other words, these pieces of evidence indicate the necessity of improving intestinal damage caused by aging to prevent various diseases.

Glasswort (*Salicornia herbacea* L.) is a halophytic plant found in tideland or seashores. It thrives in environments with high salt concentrations. It accumulates a significant amount of salt compared to land plants. Glasswort has been reported to contain high contents of dietary fibers, phytochemicals, and minerals such as magnesium, calcium, potassium, and iron [8]. Recently, an imbalanced diet has led to a deficiency in micronutrients such as iron, zinc, and minerals in a global population of two billion individuals. This micronutrient deficiency has been observed to have adverse effects on physiological functions throughout all age groups, ranging from young children to the elderly [9]. Recent studies have also shown that micronutrient deficiency can significantly impact human gut microbiota [10], contribute to DNA damage, accelerate aging, and increase cancer risk [11]. Furthermore, phytochemicals such as polyphenols and carotenoids are known to provide various physiological benefits including anti-inflammation [12], cardiovascular disease-preventing [13], and anti-cancer [14] effects. Several studies have demonstrated that phytochemicals can modulate gut microbiota compositions, thereby improving gut-related disorders such as inflammatory bowel disease (IBD) [15]. Glasswort contains dicaffeoylquinic acid derivatives and flavonoid glucoside known to exhibit high antioxidative activities [16]. Moreover, glasswort has various biological effects including antioxidant properties [17], anti-cancer activity, anti-obesity effects [18], and immunomodulatory activity [19]. Thus, glasswort could be used as a supplement for minerals and phytochemicals.

Fermented foods are associated with a wide range of health benefits. Moreover, the fermentation process can enhance the bioavailability of nutrients and phytochemicals present in these foods [20]. Fermentation process involves lysis of glycosides by microorganisms, resulting in production of aglycones. These aglycones possess diverse physiological functions. They may potentially contribute to health benefits [21]. Recent clinical studies have indicated that fermented foods can reduce the risk of inflammatory diseases and significantly impact gut microbiome compositions [22,23]. The fermentation process applied to glasswort has been found to enhance its antioxidant activity by significantly increasing levels of polyphenols and flavonoids, thus highlighting its enhanced bioactive properties [24,25]. Although previous studies have indicated potential health benefits of fermented glasswort, no experimental studies have been conducted to investigate effects of fermented glasswort on gut health in aging animal models.

Thus, the purpose of this study was to examine the improvement effect of fermented glasswort on gut health through D-galactose-induced aging animal model. To assess gut health, this study investigated histological changes such as mucosal thickness and goblet cell count in the colon known to be crucial for maintaining a healthy barrier. Additionally, biomarkers associated with SCFAs content, oxidative stress, and inflammation were analyzed. Findings of this study will provide valuable insights into the potential of fermented glasswort as a novel approach for enhancing gut health. Given the significance of gut health in various diseases, improving it can contribute to healthy aging and disease prevention. Consequently, fermented glasswort could be used as a potential health food or supplement.

## 2. Materials and Methods

### 2.1. Preparation of Fermented Glasswort

Fermented glasswort extracts were purchased from Dasarang Agricultural Association Corporation (Shinan, Republic of Korea). In brief, 20 g of glasswort was ground with 80 mL of water and filtered using Whatman No. 1 filter paper. Glasswort filtrate was inoculated with 5% *Lactobacillus acidophilus* (KCTC 10827BP) and fermented at 30 °C for 10 days. After fermentation, the fermented glasswort filtrate was stored at −20 °C until analysis.

### 2.2. Animal Study

Twelve-week-old male Sprague Dawley (SD) rats were purchased from Orient Bio Inc. (Seongnam, Gyunggi-do, Republic of Korea). Animals were housed in controlled conditions (temperature of 22 ± 1 °C, humidity of 50 ± 5%, and 12/12-h light/dark cycle).

They were provided ad libitum access to food and water and allowed to acclimate for one week prior to experiments.

Rats were randomly divided into five groups ($n = 8$/group): control (CON), control plus D-gal (CD), fructooligosaccharide (FOS), low glasswort (LGW), and high glasswort (HGW). All groups except for the CON group that was treated with saline were intraperitoneally injected with D-gal [150 mg/kg] once a day for eight weeks. The FOS group was orally administered with fructooligosaccharide at 400 mg/kg (Korea Biopharm Co., Ltd., Jincheon, Republic of Korea). LGW and HGW groups were orally administered with fermented glasswort filtrate at 10 mg/kg and 20 mg/kg, respectively. All animal experiments were approved by the Institutional Animal Care and Use Committee (IACUC) of Mokpo National University (MNU-IACUC-2020-010). Body weights were measured every week. Food intake and water intake were measured every day. Food and water intakes were expressed as average values measured over a period of 7 days (representing one week).

### 2.3. TBARS Assay

Lipid peroxidation level in the colon was assessed using the TBARS method [26]. Briefly, 20% trichloroacetic acid in 0.6 M HCl (0.5 mL) and thiobarbituric acid in 1 M NaCl (0.3 mL) were mixed with 0.1 mL of colon. The mixture was incubated at 95 °C for 20 min in a water bath. After cooling, the mixture solution was partitioned with 0.8 mL of butanol followed by centrifugation at $1500\times g$ for 10 min. Malondialdehyde content in the butanol layer was measured at 532 nm with a spectrophotometer (Hewlett Packard, CA, USA).

### 2.4. Western Blotting

Following completion of the animal experiment, colon tissue was immediately ground using liquid nitrogen. Ground colon tissue (0.15 g) was homogenized in RIPA buffer (10 nM NaF, 2 mM phenylmethyl sulfonyl fluoride, 0.1 μg/L aprotinin, 10 mM sodium pyrophosphate, 1 μg/mL pepstatin, and 1 μg/mL leupeptin). Reagents for RIPA homogenized buffer were purchased from Sigma-Aldrich Chemical (St. Louis, MO, USA). Homogenized samples were then centrifuged and resulting supernatants were collected. Protein concentrations of samples were determined using the Bradford method.

For sodium dodecyl sulfate polyacrylamide gel electrophoresis (SDS-PAGE), proteins (100 μg) were mixed with tricine sample buffer (Bio-Rad Laboratories, Inc., Hercules, CA, USA) and subjected to heat treatment before loading onto SDS-PAGE gel. Separated proteins were transferred from the gel to a polyvinylidene fluoride (PVDF) membrane using a semi-dry transfer system. Following the transfer, the PVDF membrane was blocked with Tris-buffered saline containing 0.05% Tween 20 (TBST) plus 3% bovine serum albumin (BSA). Subsequently, the membrane was incubated with primary antibodies, including anti-TNF-α (ab205587, Abcam plc., Cambridge, UK), atni-IL-1β (ab9722, Abcam plc., Cambridge, UK), anti-NF-κB (PA1-41089, Thermo Fisher Scientific Inc., Waltham, MA, USA), and anti-GAPDH (PA1-987, Thermo Fisher Scientific Inc., Waltham, MA, USA). After incubating with primary antibodies, the membrane was washed multiple times with TBST buffer. The membrane was then incubated with a secondary antibody, horseradish peroxidase (HRP)-conjugated goat anti-rabbit immunoglobulin G. Following another round of washing with TBST buffer, protein bands were visualized using an enhanced chemiluminescence (ECL) reagent (Santa Cruz Biotechnology, Dallas, TX, USA) and captured using an iBright FL1500 imaging system (Invitrogen, Waltham, MA, USA). Intensities of protein bands were quantified using ImageJ software (NIH, Bethesda, MD, USA). To ensure equal protein loading, GAPDH was utilized as a loading control.

### 2.5. Analysis of SCFAs Using GC

To quantify contents of short-chain fatty acids (SCFAs), cecum samples (1 g) from rats were mixed with methanol and homogenized for 3 min. The pH of the mixture was then adjusted to 2–3 by adding 5 M HCl. Following that, the mixture was kept at room temperature, with shaking occurring every 2 min for a total duration of 10 min. Subsequently, the mixture was centrifuged at 5000 rpm for 20 min. The resulting clear supernatant was utilized for gas chromatography (GC) analysis. For each mixture, a internal standard solution of 2-ethylbutyric acid (1 mM) in 12% formic acid was added. GC analysis was conducted using the methodology described in a previous study [27]. SCFAs in the mixture were quantified based on standard curves generated for n-butyric acid, acetic acid, and propionic acid.

### 2.6. Histology

Colon tissues were fixed with 4% paraformaldehyde for 1 day, embedded, and cut into 5 μm-thick sections on glass slides. For hematoxylin and eosin (H&E) staining, sections were de-paraffinized in xylene and dehydrated in graded alcohol. Slides were stained with hematoxylin solution for 6 min and subsequently rinsed with tap water for 5 min. Following that, slides were stained with eosin solution for 1 min. For Alcian blue (AB) staining, 5 μm-thick sections of colon tissues were immersed in acetic acid for 3 min and then dyed with AB solution for 30 s.

For measuring mucus layer thickness, one-cm segments of colon with fecal contents were carefully obtained and immersed in water-free methanol-carnoy's solution for 24 h. These tissues were washed with methanol before embedding with paraffin and cut into 5 μm-thick sections onto glass slides. Sections were de-paraffinized in xylene and dehydrated in grade alcohol. Next, slides were oxidized in 0.5% periodic acid solution for 5 min. These slides were then dyed with Schiff's reagent for 15 min.

All stained slides were dehydrated in graded alcohol and placed in xylene. Finally, slides were mounted using Canada balsam (Sigma-Aldrich Chemicals, St. Louis, MO, USA). Images were acquired at 20–40 magnification using an Olympus BX43 microscope (Olympus, Tokyo, Japan). The number of goblet cells was quantified as the average number of goblet cells within 10 crypts in each colon tissue. The thickness of the mucus layer of colon was quantified using ImageJ software.

### 2.7. Statistical Analysis

All data are presented as the mean $\pm$ standard deviation (SEM). To assess normality and homogeneity of the data, both the Shapiro–Wilk normality test and Levene's test were conducted. Comparisons between multiple groups were performed using one-way analysis of variance (ANOVA) with Tukey's test. All statistical analyses were performed using IBM SPSS 21 software and $p < 0.05$ was considered to be statistically significant.

## 3. Results

### 3.1. General Characteristics

During the experiment, body weight, food intake, and water intake of all groups were monitored. Body weights of all groups increased over time, with the FOS group showing the highest increase among all groups. The HGW group appeared to have the lowest body weight during the experiment, although there was no significant difference in body weight among groups (Figure 1A). For food and water intakes, the LGW group had higher intakes than the HGW group (Figure 1B,C). The HGW group tended to have lower food and water intakes during the experiment.

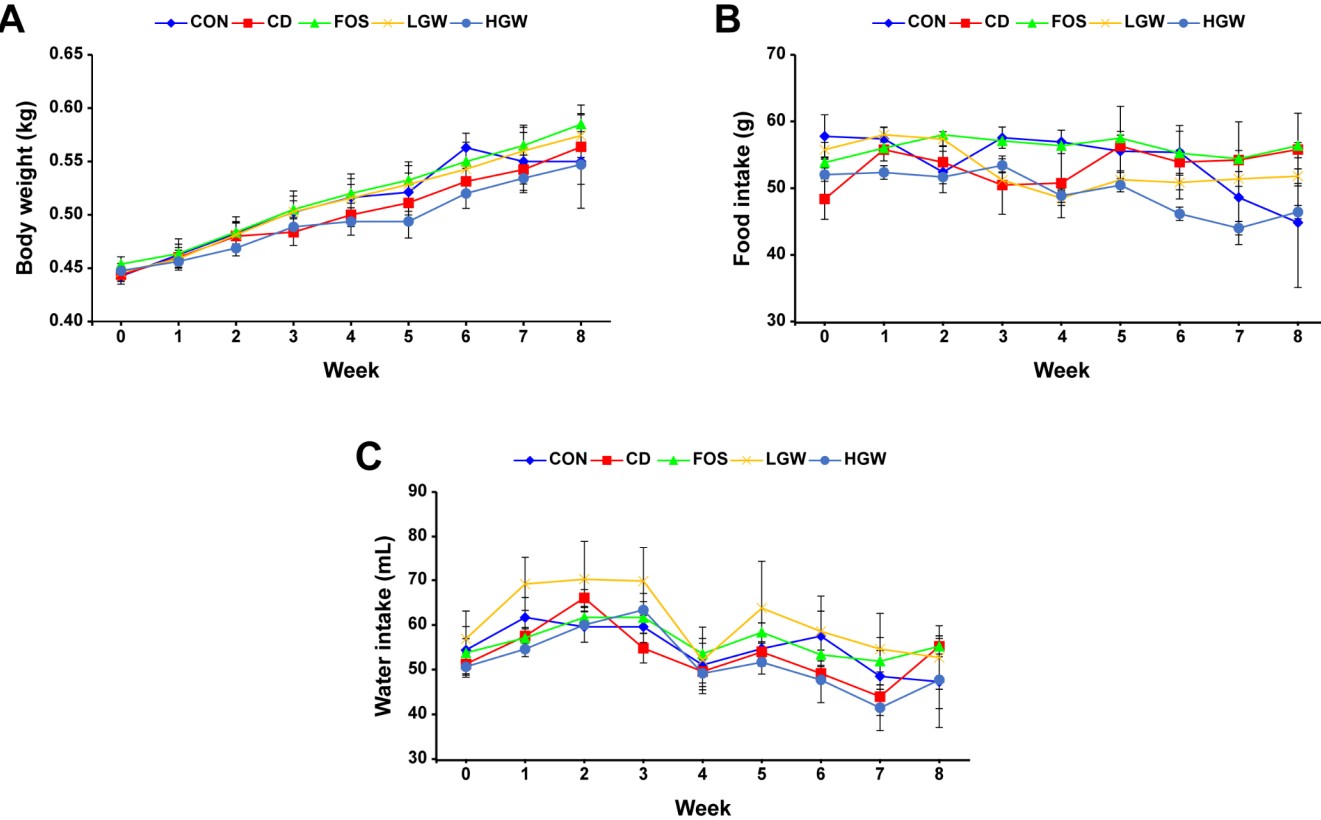

**Figure 1.** General characteristics of D-galactose-induced aging rats. (**A**) Body weight. (**B**) Food intake. (**C**) Water intake. Rats were orally administered with different treatment and subjected to continuous observation and monitoring over an 8-week period. CON, control; CD, control plus D-gal; FOS, fructooligosaccharide plus D-gal; LGW, low glasswort plus D-gal; HGW, high glasswort plus D-gal. Values are presented as the means ± SEM for eight rats in each group.

### 3.2. Histopathology and Number of Goblet Cells

Histopathological evaluation of the colon was performed by performing H&E staining for colon sections. Histological damage of the colon tissue was assessed using a previously established protocol [28]. The CD group exhibited crypt loss and irregular crypts such as non-parallel crypts and variable crypt diameters compared to the CON group (Figure 2A). The FOS group as a positive control group showed a slight improvement in colon damage, while groups fed with fermented glasswort appeared to show improvement of crypt damage compared with the CD group.

To detect goblet cells known to secrete mucins, colons were subjected to AB staining for both neutral and acidic mucins (Figure 2B,C). The CD group showed a significantly lower number of goblet cells than the CON group ($p < 0.05$). The number of goblet cells was significantly increased in the HGW group compared to that in the CD group. It appeared to increase in FOS and LGW groups ($p < 0.05$), although statistically significant differences were not observed among CD, FOS, and LGW groups. Notably, the number of goblet cells in the HGW group was similar to that in the CON group.

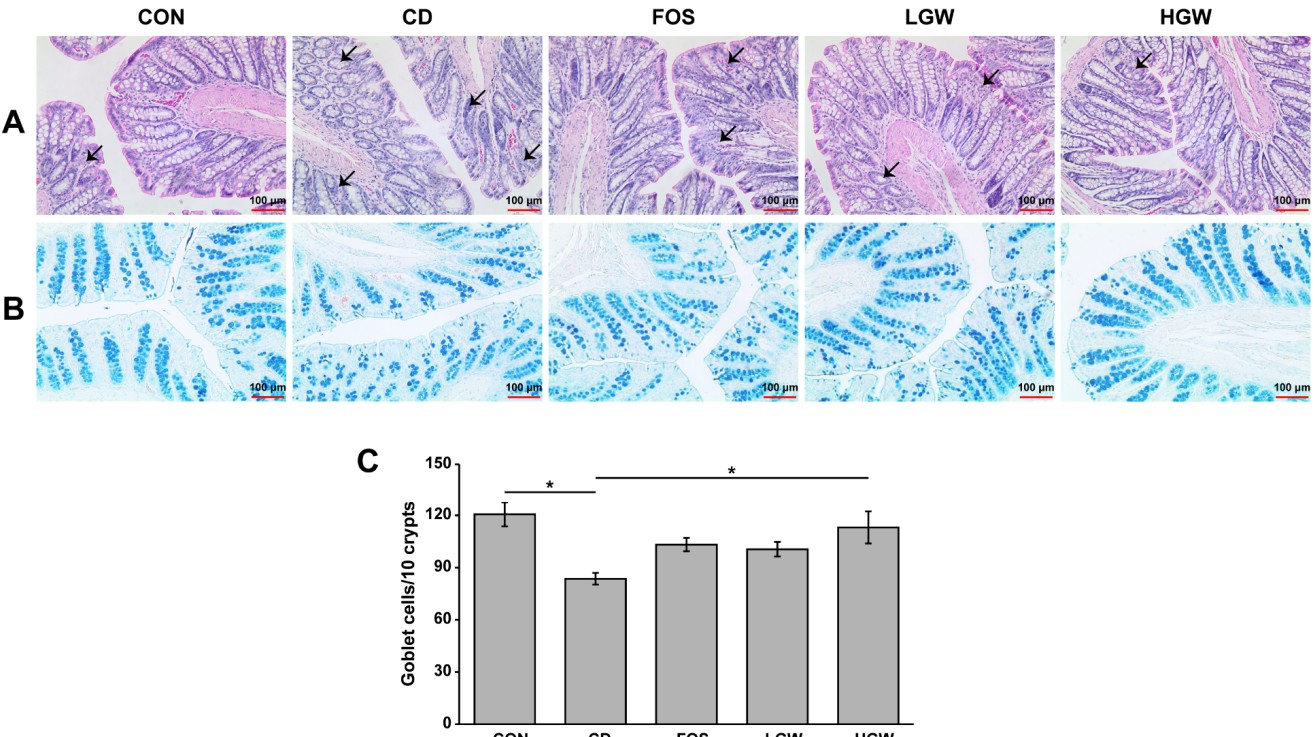

**Figure 2.** Histological evaluation of colons of D-galactose-induced aging rats. (**A**) Representative images of H&E-stained colon tissues. Arrows indicate the presence of crypt damage, villous atrophy, and a reduction in goblet cells. (**B**) Representative images of Alcian blue-stained colon tissues. Blue color represents goblet cells. (**C**) Quantification of goblet cells in Alcian blue-stained colon tissues. CON, control; CD, control plus D-gal; FOS, fructooligosaccharide plus D-gal; LGW, low glasswort plus D-gal; HGW, high glasswort plus D-gal. Values are presented as the means $\pm$ SEM for eight rats in each group. Statistically significant difference (* $p < 0.05$) was determined by one-way ANOVA followed by Tukey's test for multiple comparisons. Original magnification, 40×.

### 3.3. Mucus Layer Thickness of Colon

The mucus layer of the colon was subjected to periodic acid-Schiff (PAS) staining and the mucus layer thickness was measured (Figure 3). The CD group (19.99 $\pm$ 2.49 μm) showed a significant decrease in mucus layer thickness than the CON group (33.46 $\pm$ 2.23 μm, $p < 0.05$), indicating that D-gal-induced aging decreased the thickness of mucus layer. Interestingly, the mucus layer thickness of the HGW group (33.62 $\pm$ 2.08 μm) recovered to the level of the CON group. Mucus layer thicknesses of colons for FOS and LGW groups were 23.82 $\pm$ 2.70 μm and 31.06 $\pm$ 2.23 μm, respectively.

### 3.4. Short-Chain Fatty Acids (SCFAs)

Levels of SCFAs in cecal contents were analyzed (Figure 4). The HGW group exhibited a significantly increased level of butyric acid which was decreased in CD and FOS groups ($p < 0.05$). There was no statistically significant difference in acetic acid or propionic acid level among groups. Trends of changes in butyrate levels were similar to changes in goblet cell number.

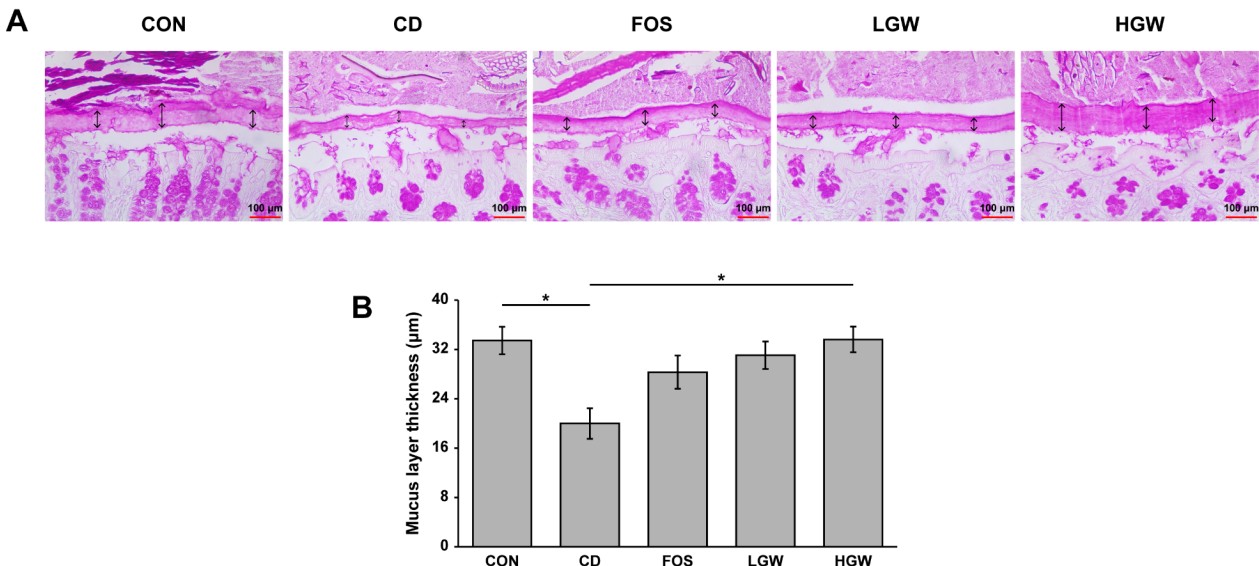

**Figure 3.** Colon mucus layer thicknesses of D-galactose-induced aging rats. (**A**) Representative images of PAS-stained colon tissues. Mucus layer thickness (μm) increased in rats supplemented with fermented glasswort extract. (**B**) Quantification of mucus layer thickness of PAS-stained colon tissues. CON, control; CD, control plus D-gal; FOS, fructooligosaccharide plus D-gal; LGW, low glasswort plus D-gal; HGW, high glasswort plus D-gal. Dark blue color indicates goblet cells. Values are presented as the means ± SEM for eight rats in each group. Statistically significant difference (* $p < 0.05$) was determined by one-way ANOVA followed by Tukey's test for multiple comparisons. Original magnification, 40×.

### 3.5. Oxidative Stress-Related Markers

To determine the effect of fermented glasswort on oxidative stress, we measured MDA levels in colons. TBARS assay is a well-known method for measuring lipid peroxidation in tissues. Compared to the CON group, the CD group had significantly increased MDA levels, suggesting accumulation of D-gal-induced cellular oxidative stress (Figure 5). MDA levels in LGW and HGW groups were decreased compared to those in the CD group while MDA levels in the FOS group were similar to those of the CD group. Especially, MDA levels in the HGW group were significantly decreased compared to those in FOS and CD groups ($p < 0.05$).

### 3.6. Expression Levels of Inflammation-Related Proteins

The aging process is associated with an increased production of reactive oxygen species (ROS), which subsequently triggers an inflammatory response [29]. Thus, protein expression levels of inflammation-related markers such as TNF-α, NF-κB, and IL-1β were analyzed. Expression levels of TNF-α appeared to decrease slightly in FOS and HGW groups than in the CD group, although significant difference was not observed (Figure 6A). Expression patterns of NF-κB were similar to those of IL-1β (Figure 6B,C). Significant difference was not observed among groups. Interestingly, fermented glasswort groups (LGW and HGW) exhibited significantly decreased expression levels of IL-1β than the CD group ($p < 0.05$). Indeed, the CD group showed higher expression level of IL-1β than the CON group (Figure 6B, $p < 0.05$), suggesting that inflammation was increased by D-gal treatment.

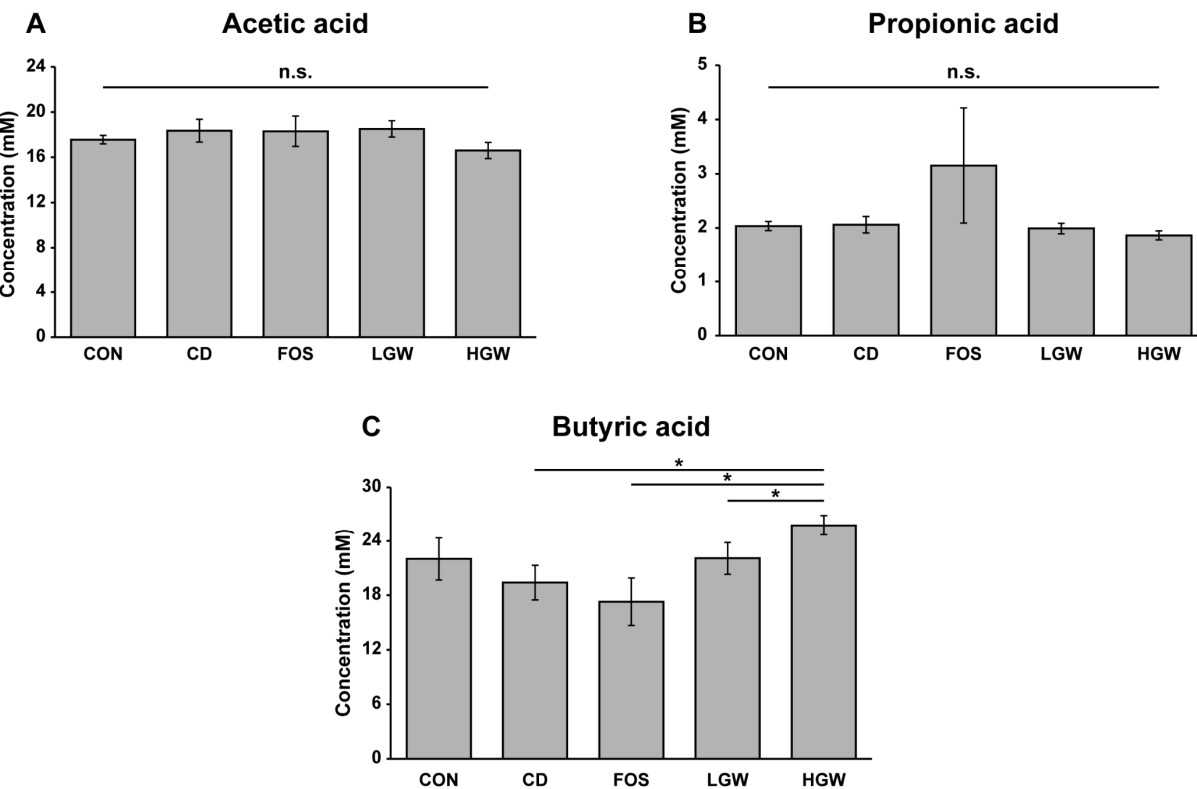

**Figure 4.** Levels of SCFAs in D-galactose-induced aging rats. Levels of acetic acid (**A**), propionic acid (**B**), and butyric acid (**C**) in cecal contents were analyzed by GC. CD, control plus D-gal; FOS, fructooligosaccharide plus D-gal; LGW, low glasswort plus D-gal; HGW, high glasswort plus D-gal. Values are presented as the means ± SEM for eight rats in each group. Statistically significant difference (* $p < 0.05$) was determined by one-way ANOVA followed by Tukey's test for multiple comparisons. n.s., not significant.

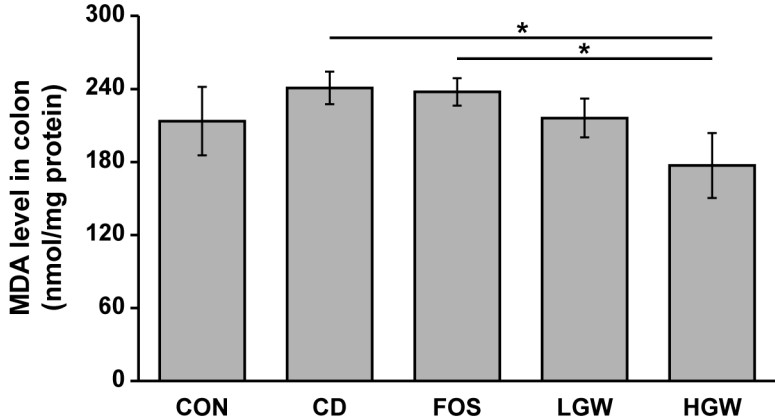

**Figure 5.** MDA levels in D-galactose-induced aging rats. CON, control; CD, control plus D-gal; FOS, fructooligosaccharide plus D-gal; LGW, low glasswort plus D-gal; HGW, high glasswort plus D-gal. Values are presented as the means ± SEM for eight rats in each group. Statistically significant difference (* $p < 0.05$) was determined by one-way ANOVA followed by Tukey's test for multiple comparisons.

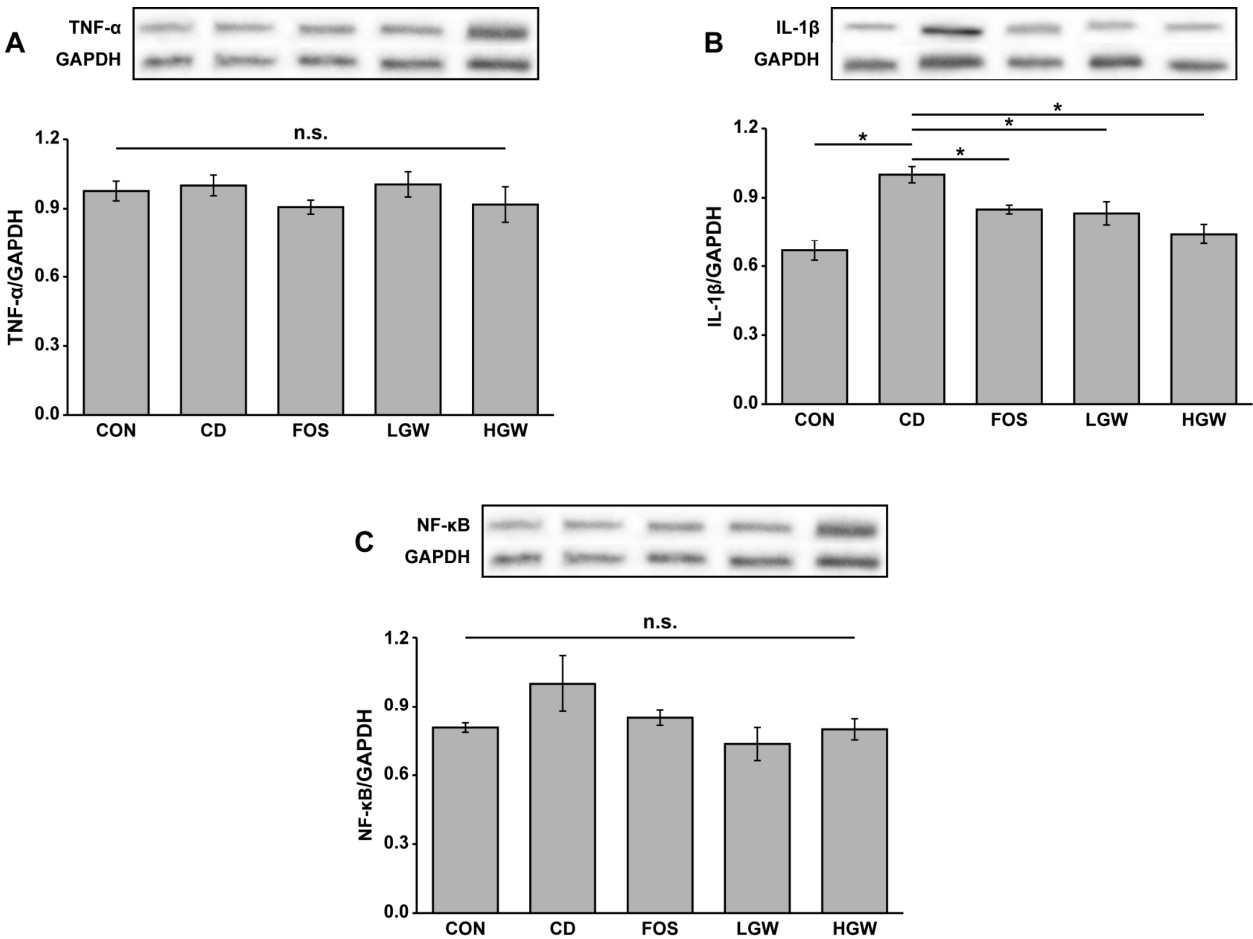

**Figure 6.** Relative protein expression levels of TNF-α (**A**), IL-1β (**B**), and NF-κB (**C**) in colons of D-galactose-induced aging rats. CON, control; CD, control plus D-gal; FOS, fructooligosaccharide plus D-gal; LGW, low glasswort plus D-gal; HGW, high glasswort plus D-gal. Values are presented as the means ± SEM for eight rats in each group. Statistically significant difference (* $p < 0.05$) was determined by one-way ANOVA followed by Tukey's test for multiple comparisons. n.s., not significant.

## 4. Discussion

In recent years, the prevalence of aging-related disorders has become a significant challenge due to the increasing population of elderly individuals. Westernized dietary habits have contributed a large part to the rapid increasing numbers of intestinal diseases such as colon cancer, leaky gut syndrome, and IBD [30]. Accordingly, the food industry has begun to discover food materials that can maintain health and treat or prevent various diseases, with increasing interest in gut health, especially in aging-related health. Glasswort contains a variety of physiologically active compounds, including minerals, flavonoids, and phytochemicals [8]. Fermentation of plants offers numerous advantages, including enhanced physiological functional components and improved physical properties [20]. In this study, we examined the impact of fermented glasswort on an animal model of D-gal-induced aging, aiming to elucidate its effects and potential therapeutic applications.

D-gal has been commonly used to induce aging in animal models. Excessive intake of D-gal can lead to the production of galactitol through aldose reductase in the body. Galactitol is a toxic metabolite that can increase reactive oxygen species (ROS) production, reduce activation of antioxidant enzymes, and form various oxidized substances. These factors contribute to organ damage and accelerate the aging process [31]. Our study demonstrated that D-gal-induced histological damage of colon was improved in fermented glasswort diet groups such as LGW and HGW groups. H&E staining results revealed

considerable crypt damage in the CD group, whereas fermented glasswort groups exhibited less pronounced crypt damage. Previous studies have reported that aged mice show significantly increased inflammatory cell infiltrate, depletion of goblet cells, and crypt abscesses in colon compared to adult mice [32]. Stem cells in the gut are present at the base of the colonic crypt. They produce various cells. However, with aging, the function of these stem cells declines in tissues such as the skin, brain, bone, and gut. Age-related disorders are caused by stem cells exhaustion [33]. Four different types of intestinal epithelial cells, columnar cells, goblet cells, neuroendocrine cells, and Paneth cells, are produced by intestinal stem cells [34]. Goblet cells can secrete glycoprotein mucin, a major component of the intestinal mucus. Mucus plays an important role in protecting the gastrointestinal tract. It has been reported that the two mucus layers of the colon are constituted by MUC2 mucin [35]. Continuous damage to the intestine can result in destruction of crypts known to protect intestinal stem cells, leading to a decrease in goblet cell number. Results of staining colon tissues shown in Figure 2 revealed that goblet cells damage was effectively prevented in fermented glasswort groups such as LGW and HGW groups where the CD group exhibited depletion of goblet cells. In addition, trends of goblet cell numbers showed a similar pattern to changes observed for mucus layer thickness (Figure 3). Recent studies have shown that aging is associated with increased apoptosis of goblet cells, reduced abundance of beneficial bacteria such as *Akkermansia* spp., and decreased expression of genes related to immune function [36]. Excessive D-galactose intake (800 mg/kg in mice) has been shown to decrease expression levels of tight junction proteins, including occludin and claudin-1, which are crucial for maintaining the integrity of the intestinal barrier, and disrupt the balance of gut microbiota [37]. Furthermore, D-gal administration (150 mg/kg in mice) can lead to an imbalance in gut microbiota, resulting in a decrease in the production of important metabolites such as SCFA [38]. Further studies are necessary to elucidate whether the reduction in goblet cell number in the CD group is due to the damage of intestinal stem cells or other factors. In a previous study, we found that fermented glasswort contained a significant amount of minerals and free amino acids such as alanine, proline, aspartic acid, and lysine [17]. In addition, the fermentation process can significantly influence metabolic changes in glasswort compared to non-fermented glasswort [39]. Among these metabolites, phenolic compounds are recognized for their ability to scavenge free radicals, thereby functioning as antioxidants and exhibiting anti-inflammatory properties [20]. Numerous studies have demonstrated that phytochemicals can influence compositions of intestinal microflora and reduce intestinal permeability by preserving the integrity of the mucous layer [40]. Notably, compositions of intestinal microbiota in elderly individuals have been found to be influenced by diet, which in turn can affect their health status [41]. Many studies have revealed that butyrate is involved in mucin production by regulating MUC genes expression in intestinal epithelial goblet cells [42]. Our results revealed that trends of mucus layer thickness changes had a similar pattern to changes observed for butyric acid levels. Using an aging animal model, we demonstrated that consumption of fermented glasswort diet could enhance the production of butyric acid, a metabolite of intestinal microorganisms, ultimately improving barrier function of the intestine. This implies that fermented glasswort might have a beneficial effect on intestinal health in elderly individuals. Based on these findings, future studies should focus on profiling intestinal microbiota and investigating the underlying mechanisms associated with gut health through a fermented glasswort diet.

Reactive oxygen species (ROS) have high reactivity and give damage to cellular components such as nucleic acid, lipids, and proteins. Oxidative stress induced by ROS can trigger inflammatory responses and accelerate the aging process, thereby playing a significant role in the development of diseases [29]. Under a normal physiological state, homeostasis of ROS is regulated by biological processes. Aging is characterized by chronic inflammation known as immunosenescence [43]. Inflammation is a physiological defense system that can repair tissues when they are damaged. It involves immune cells and molecular mediators. Therefore, there is a persistent effort in the development of food materials that demonstrate

antioxidant properties. In our study, the fermented glasswort diet resulted in a reduction in the content of MDA, a marker of oxidative stress. Additionally, it effectively decreased protein expression levels of IL-1β and NF-κB. In an animal model with adenine-induced kidney damage, administration of fermented glasswort extract resulted in reduced levels of blood urea nitrogen (BUN) and creatinine, both of which are indicators associated with oxidative stress [44]. Main phenolic compounds identified in glasswort were vanillic acid and *p*-coumaric acid. These compounds exhibit anti-cancer effects in colon cancer cells [45]. These effects might be attributed to antioxidant and anti-inflammatory properties conferred by high mineral contents and phytochemical compounds present in fermented glasswort. Findings of this study suggest that fermented glasswort might possess antioxidant and anti-inflammatory properties, which could be valuable for managing oxidative stress-related conditions and chronic inflammation associated with aging.

FOS is a well-known prebiotic that can enhance beneficial bacteria, reduce inflammation, and contribute to improved gut health [46]. Therefore, FOS was employed as a positive control in this study. The FOS group was treated with FOS at a dose equivalent to 400 mg FOS/kg rat, which corresponded to approximately 4 g FOS/60 kg human when calculated as described elsewhere [47]. The FOS group showed improvements in crypt appearance, goblet cell number, mucus layer thickness, and expression of inflammation-related biomarkers, similar to improvements in the LGW group. The LGW group was treated with 10 mg fermented glasswort filtrate in dry weight/kg rat, which was approximately equivalent to 4 g fresh glasswort in dry weight before fermentation/60 kg human. Notably, the level of MDA in the FOS group was similar to that in the CD group, suggesting that FOS might not be as effective as fermented glasswort for improving oxidative stress in aged gut. These findings suggest that fermented glasswort could be at least as effective as FOS for improving aged gut health.

## 5. Conclusions

Consumption of fermented glasswort appeared to reduce intestinal oxidative stress and inflammation in a D-gal-induced aging animal model. Furthermore, production of butyrate, number of goblet cells, and thickness of mucus layer in colons were increased by consumption of fermented glasswort. These findings indicate that fermented glasswort has potential to positively improve intestinal health in the elderly population (Figure 7). Furthermore, these results suggest the potential utilization of fermented glasswort as a prebiotic for enhancing intestinal health of the elderly population.

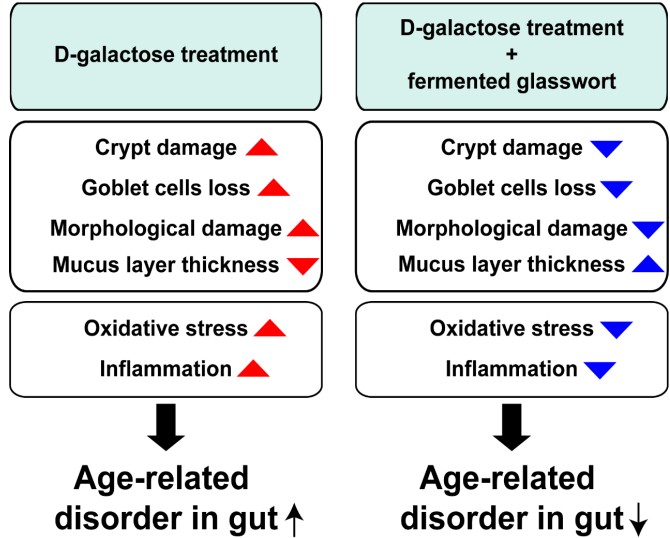

**Figure 7.** A schematic diagram illustrating effects of fermented glasswort on aged gut induced by D-galactose.

**Author Contributions:** Conceptualization, K.-S.H., D.S. and N.S.; methodology, D.S. and N.S.; software, D.S. and N.S.; validation, D.S. and N.S.; formal analysis, D.S., N.S. and S.C.; investigation, D.S., N.S. and S.C.; resources, Y.-K.P.; data curation, D.S.; writing—original draft preparation, D.S. and N.S.; writing—review and editing, K.-S.H., D.S. and N.S.; visualization, D.S.; supervision, K.-S.H.; project administration, D.S.; funding acquisition, K.-S.H. and Y.-K.P. All authors have read and agreed to the published version of the manuscript.

**Funding:** This study was supported by the "Leaders in Industry-university cooperation +" Project funded by the Ministry of Education and National Research Foundation, Republic of Korea.

**Institutional Review Board Statement:** The animal study protocol was approved by the Institutional Animal Care and Use Committee (IACUC) of Mokpo National University, Jeonnam, Korea (approval number: MNU-IACUC-2020-010).

**Informed Consent Statement:** Not applicable.

**Data Availability Statement:** Not applicable.

**Conflicts of Interest:** The authors have no conflict of interest relevant to this study to disclose.

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
