# Peer review of "Protective Effects of Fermented Glasswort (Salicornia herbacea L.) on Aged Gut Induced by D-Galactose in Rats"

_applsci, doi:10.3390/app13148386_

Round 1

Reviewer 1 Report

In this manuscript, the author summarized and discussed the effect of fermented glasswort on gut health in D-galactose (D-gal)-induced aging rats. It is also confirmed that fermented glasswort ameliorates age-related gut health. On this basis, the author further confirmed that consumption of fermented glasswort seems to reduce intestinal oxidative stress and inflammation. In addition, production of butyrate, number of goblet cells, and thickness of mucus layer in colon were increased by consumption of fermented glasswort. In general, fermented glasswort could be used to improve aging-related disorders in gut. The study contributes to the understanding of the potential health benefits of glasswort for gut health. The manuscript is well organized and clearly expressed. I suggest solving the following problems in major revision:

1. The sentence patterns of juxtaposed sentences with “and” can be reduced as much as possible.

2. Some references are too old.

3. Some paragraphs in the introduction are too short, so it is suggested to plan and arrange them reasonably, and naturally express them excessively. The AGEs related index should be measured (Critical Reviews in Food Science and Nutrition. Doi: 10.1080/10408398.2022.2076064.).

4. The introduction should include the main contents and conclusions of this paper, with clear logic and appropriate details.

5. There is a problem that the first letter is not capitalized in line 131, so it is suggested to optimize it.

6. In this manuscript, the color of the error line of the graph is the same as that of the histogram, which can't be clearly distinguished, so it is suggested to optimize it.

7. There is a sentence syntax error in line 225, which is suggested to be revised.

8. The figures can be further optimized to make it more beautiful.

9.The gut microbiota structure should be measured (Food Bioscience. 50(2022): 101946.).

10.The future prospect study should be added.

11. It is suggested that the conclusions should be separated separately, and the beneficial application of Fermented Glasswort to be developed should be added to enrich the conclusion.

12. please update the reference in recent years.

In this manuscript, the author summarized and discussed the effect of fermented glasswort on gut health in D-galactose (D-gal)-induced aging rats. It is also confirmed that fermented glasswort ameliorates age-related gut health. On this basis, the author further confirmed that consumption of fermented glasswort seems to reduce intestinal oxidative stress and inflammation. In addition, production of butyrate, number of goblet cells, and thickness of mucus layer in colon were increased by consumption of fermented glasswort. In general, fermented glasswort could be used to improve aging-related disorders in gut. The study contributes to the understanding of the potential health benefits of glasswort for gut health. The manuscript is well organized and clearly expressed. I suggest solving the following problems in major revision:

1. The sentence patterns of juxtaposed sentences with “and” can be reduced as much as possible.

2. Some references are too old.

3. Some paragraphs in the introduction are too short, so it is suggested to plan and arrange them reasonably, and naturally express them excessively. The AGEs related index should be measured (Critical Reviews in Food Science and Nutrition. Doi: 10.1080/10408398.2022.2076064.).

4. The introduction should include the main contents and conclusions of this paper, with clear logic and appropriate details.

5. There is a problem that the first letter is not capitalized in line 131, so it is suggested to optimize it.

6. In this manuscript, the color of the error line of the graph is the same as that of the histogram, which can't be clearly distinguished, so it is suggested to optimize it.

7. There is a sentence syntax error in line 225, which is suggested to be revised.

8. The figures can be further optimized to make it more beautiful.

9.The gut microbiota structure should be measured (Food Bioscience. 50(2022): 101946.).

10.The future prospect study should be added.

11. It is suggested that the conclusions should be separated separately, and the beneficial application of Fermented Glasswort to be developed should be added to enrich the conclusion.

12. please update the reference in recent years.

Reviewer 2 Report

Song et al. explore the benefits of the Gasswort plant, which according to the literature contains various bioactive compounds that can benefit the health of the host. To demonstrate the benefits, the authors performed a mouse model of aging. For this purpose, Song et al. carry out  an aged in vivo model of  induced by D-galactose. The benefits of Gasswort plant can be observed in some markers of intestinal permeability, oxidative stress and inflammation in the colon. However, significant changes must be made. There are experimental discrepancies. The association between the fermented plant and specific parameters of the anti-aging effect is not resolved.  In this work, only measurements of local oxidative stress, inflammation, and permeability (colon) is presented and not the neurological or systemic level is performed. For the evaluation of anti-aging in the proposed model, memory improvements and neurological measurements should be evaluated. Oxidative stress and inflammation are not specific risks of aging, but also other conditions, as is seen in the lines 268-278 (obesity, IBD).

The results also suggest that the D-galactose-induced aging model may not have been fully optimised. For example, in the SFCAs content, even if the fermented plant had a significant increase compared to the CD group, there is no difference with the untreated group versus CD group. The same applies to MDA levels in the colon.

Therefore, I propose that the animal experiment not be presented as aged model, but rather as a model for altering intestinal permeability. For example, the work titled: D-Galactose Induces Chronic Oxidative Stress and Alters Gut Microbiota in Weaned Piglets’. https://doi.org/10.3389/fphys.2021.634283

Specific comments:

Line 20: No abbreviation

Line 73: An extra group with Glasswort no fermented treatment could be useful to demonstrate the benefit of the fermentation.

Line 75: code strain name and origin

Line 96: liver or colon?

Line 103-105, 121: add provider and country

Generally it is highly recommended to add the value of 'p value' when the significant term is mentioned, p < or > 0.05. Many of the claims made are dismissed by statistical analysis and are not cited. See lines 197, 198, 224, 225, 254,

Line 161: Tukey instead turkey

Line 162: p value in lower case

Line 202-205: This should be a part of introduction section

Line 235-238: Should be a part of the discussion section.

Line 268-277: Missing references.

Line 278: Define the specific damage in the colon, because only the thickness or goblet cells were reported and no other damage like neutrophil infiltration was observed.

Line 306-307: Our results showed that mucus layer thickness is correlated with the levels of butyric acid in fermented glasswort-fed groups. Is there any statistical analysis in support of that assessment? Rho Spearman, R Pearson?

Line 324-326: Why  FOS did not enhanced health in rats in your model?

Reviewer 3 Report

In this manuscript, the authors evaluated the effect of fermented glasswort on gut health in D-galactose-induced aging rats. The authors showed that dietary fermented glasswort improved gut health in D-galactose-injected rats in several ways including increased butyrate acid, increased number of goblet cells, increased thickness of mucus layer, reduced level of TBARS, and reduced level of colon inflammation. These are novel findings. However, there are some problems: 

1. The title can be modified. D-galactose has been a popular aging model for brain. However, this paper focuses on the effect of D-gal on aged-gut. Therefore, “Protective effect of fermented glasswort (Salicornia herbacea L.) on aged gut induced by D-galactose” may be clearer.

2. In “Introduction” section, please add more detail about the nutrients difference between unfermented glasswort and fermented glasswort. More should be added about the D-galactose-induced aging model in brain and why choose this model for studying gut health.

3. In Materials and Methods, line 77, the nutrients content in fermented glasswort should be listed. In line 87, 88, please add “orally” before “administered”.

4. In Results section, in all graphics, please place scale bars in all photos.

The writing can be improved. Moderate editing is necessary to improve the manuscript.

Reviewer 4 Report

This work investigated the effects of fermented glasswort (Salicornia herbacea L.) on gut health in D-galactose-induced aging rats. There are some issues in this manuscript that should be modified as follows:

·         Abstract:

- The total number of rats used in the present study should be mentioned in the abstract.

- Page 1 Line 17: The word “intraperitoneal” should be changed to " intraperitoneally".

- The meaning of the abbreviations should be clearly defined at their first mention; e.g. TBARS.

·         Introduction:

-       The novel points in this study should be clearly addressed in the “Introduction” section.

-       Page 2 Line 55: The authors mentioned a recent study (Reference 15) that was published in 2016 which should not be considered recent. Please, revise.

·         Materials and methods:

1.    A reference for the method of preparation of the fermented glasswort should be added.

2.    The exact source, concentrations and the catalogue numbers of the used kits and chemicals should be mentioned.

3.    How did you know that the animals were acclimatized?

4.    References for the used doses of D-galactose, fructooligosaccharide, and the glasswort and duration of treatment should be added.

5.    It is advisable to measure the oxidative stress parameters other than TBARS including the antioxidant enzymes activities in the present study.

6.    How did the authors quantified the histopathological changes in the colonic tissue specimens? Was there a specific histopathological score?.

7.     I think that the histopathological examination is not sufficient. I suggest to carry out electron microscopic study of the colonic tissues because the electron microscopic changes in these tissues precede the gross changes in the histopathological examination.

8.    More details about the statistical analysis including the method of assessment of the normal distribution of data should be mentioned.

·         Results:

  1. Figure 1: Although the authors indicated in the "Methods" section that food intake and water intake were measured every day, they appeared in figure 1 to be measured every week. Please, revise.
  2. Figure 2: The legend of this figure should present a more detailed description of the histopathological findings together with addition of arrows that refer to the positive findings. Also, scale bars should be added to figures 2 and 3.
  3. A collective diagram summarizing the main findings of this study is recommended.

·         Discussion:

-       Lines 268-277: The first paragraph in the discussion has no references. Please, revise.

-       The discussion should be re-written to focus on analysis of the obtained results in view of the other studies and not to be just a mere repetition of the "Results" section.

·         Conclusion:

The possible clinical implications of the data obtained from the present study should be added to the "Conclusion".

·         General comments:  

1. The manuscript should be revised by English-naïve speaker to improve the quality of the language.

2. The manuscript should be checked regarding the grammatical errors and plagiarism.

Moderate editing of English language is required

Round 2

Reviewer 1 Report

The author did not follow the reviewer's suggestion and update the referred reference. Please verify those changes.

The author did not follow the reviewer's suggestion and update the referred reference. Please verify those changes.

Reviewer 2 Report

Thank you for taking those considerations into account. I have no further comments to make. I only consider that the manuscript should be reviewed by an English native speaker.

Author Response

Thank you for your comments. We received English editing services from professional editing company.

Reviewer 4 Report

The authors had appropriately addressed most of my comments

Minor editing of the English language is required

Author Response

(The authors gave the same response as above.)
